# Bayesian Spillover Graphs for Dynamic Networks

**Grace Deng**[1]  **David S. Matteson**[1]

[1]Department of Statistics & Data Science, Cornell University, Ithaca, NY, USA

## Abstract

We present Bayesian Spillover Graphs (BSG), a novel method for learning temporal relationships, identifying critical nodes, and quantifying uncertainty for multi-horizon spillover effects in a dynamic system. BSG leverages both an interpretable framework via forecast error variance decompositions (FEVD) and comprehensive uncertainty quantification via Bayesian time series models to contextualize temporal relationships in terms of systemic risk and prediction variability. Forecast horizon hyperparameter $h$ allows for learning both short-term and equilibrium state network behaviors. Experiments for identifying source and sink nodes under various graph and error specifications show significant performance gains against state-of-the-art Bayesian Networks and deep-learning baselines. Applications to real-world systems also showcase BSG as an exploratory analysis tool for uncovering indirect spillovers and quantifying systemic risk.

## 1 INTRODUCTION

We consider the task of learning temporal interactions and important components over time in a dynamic network. Many real-world systems can be described by a multivariate time series (MTS) and a natural framework for analyzing temporal relationships is Granger causality [Granger, 1969], which tests for whether one time series is useful for forecasting another one. Network Granger causality (NGC) [Basu et al., 2015] extends this concept into the multivariate setting. NGC is useful for identifying one-step ahead predictive relationships within a system, and may be considered causal under very specific conditions [Pearl et al., 2000].

Many methods have been developed to estimate NGC. Vector Autoregression (VAR) [Sims, 1980] and its variants [Lütkepohl, 2005] remain a standard-bearer for macroeconomics and financial forecasting. Bayesian networks [Pearl, 2011; Ben-Gal, 2008] are also a powerful collection of probabilistic graph models for learning NGC, usually via a directed acyclic graph (DAG). Dynamic Bayesian Networks (DBN) [Murphy, 2002] are particularly useful for modeling state changes and temporal structure learning, although it is restricted by acyclic representations. Alternative methods for estimating NGC adjacency matrices use deep learning variants, e.g., attention networks [Nauta et al., 2019], Statistical Recurrent Units (SRU) [Khanna and Tan, 2019], and sparse RNNs [Tank et al., 2018]. Recently, Generalized Vector Autoregression (GVAR) [Marcinkevičs and Vogt, 2021], which utilizes Self-explaining Neural Nets (SENN), also proposed aggregating model coefficients over lagged time series to estimate signs of NGC in addition to edge detection.

However, NGC has several drawbacks. First, it is not designed to capture cumulative interactions or multi-step ahead effects that evolve over longer forecast horizons [Marcinkevičs and Vogt, 2021], which may be particularly important in forecasting or inference for real-world systems [Diebold and Yılmaz, 2014; Billio et al., 2012]. Spillovers, in particular, is an interesting subset of temporal relationships (graph edges) that can materialize beyond 1-step ahead forecasts [Diebold and Yilmaz, 2015] in the context of forecast variability and network connectivity. Furthermore, indirect spillovers between components can also manifest via intermediary nodes despite having no direct link via NGC. Estimating NGC via DAG constraints are hence not representative of true network interactions, which can be self-directed, bi-directional, or cyclic over time. Prior NGC methods also do not quantify strengths of temporal relationships [Marcinkevičs and Vogt, 2021] nor provide ample interpretation for related graph measures. Identification of important nodes relies on standard graph theory metrics [Kramer et al., 2009; Yusoff and Sharif, 2016] such as eigen-centrality [Bonacich, 1987] or in/out degrees [Freeman, 1978]. These metrics are also static point estimates based on NGC graphs.

*Accepted for the 38th Conference on Uncertainty in Artificial Intelligence* (UAI 2022).

And although methods such as GVAR offer sign estimation for temporal relationships, the actual coefficient values (edge weights) are not necessarily meaningful.

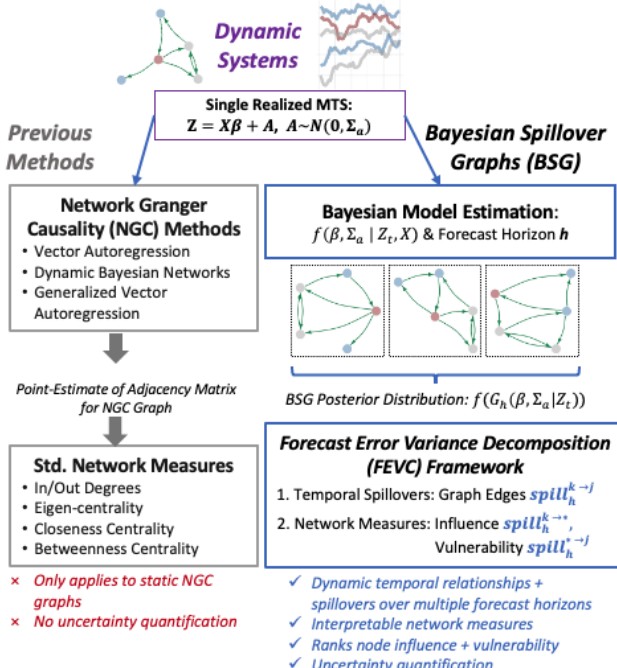

Figure 1: Comparison of BSG vs. Prior NGC Methods. BSG combines Bayesian VAR estimation with interpretable FEVD framework over forecast horizons $h$ to quantify strength of temporal interactions (BSG edge weights) and systemically important nodes over time.

To summarize, the major drawbacks of current methods are (1) lack of flexibility for observing network interactions over multiple forecast horizons, (2) lack of interpretable network measures that are contextualized, (3) and lack of uncertainty quantification for strength of temporal relationships and node influence. To this end, a promising solution is to leverage forecast error variance decomposition (FEVD) from classic time series forecasting, which estimates the temporal effect of shocks to individual nodes in the system [Barbaglia et al., 2020; Tsay, 2013; Diebold and Yilmaz, 2015], and Bayesian VAR models [Rossi et al., 2012; Koop and Korobilis, 2010] which provide comprehensive uncertainty quantification.

In particular, the formulae behind FEVD is a cornerstone of classic multivariate time series analysis when we are interested in relationships between time series components. It is commonly cited as (generalized) impulse response functions in statistical literature and multiplier analysis in economic literature [Tsay, 2013], and key applications include quantifying the effect of one time series component over forecast horizons, a key advantage over NGC. Under careful assumptions and conditions, it can also be a viable causal inference tool to analyze impact of specific policies [Swan-

son and Granger, 1997]. The idea of standardizing FEVD as a measure of risk and connectivity has been motivated by macroeconomic and financial applications [Diebold and Yilmaz, 2015; Barbaglia et al., 2020].

Formally, we define spillovers as the predicted impact of one component on all other components in a dynamic network with respect to forecast variability and forecast horizon $h$. Intuitively, we are learning how unexpected shocks in one component cascades throughout the network to all other components, as well as examining how this impact evolves over time. Statistically, we can estimate $h$-step ahead spillovers based on normalized FEVD for one-step ahead forecasts and beyond after parameter estimation via Bayesian VAR; interpretation of resulting spillover effects is then contextualized by the input time series while also accounting for parameter estimation variability.

**Motivation.** We present Bayesian Spillover Graph (BSG) for analyzing temporal interactions over multiple forecast horizons, identification of systemic influential and at-risk nodes, and uncertainty quantification for novel network measures with interpretation beyond simple NGC. BSG is both a powerful exploratory data analysis and inference tool; key contributions include:

1. We model temporal relationships in a dynamic system based on a single observed MTS; forecast horizon hyperparameter $h$ allows for flexibility in learning short-term vs. long-term spillover effects.
2. We propose interpretable network measures for contextualizing spillovers with respect to prediction variability and identifying sink and source nodes within a dynamic network. We demonstrate the robustness of these measures across various graph and error dependency specifications.
3. We provide uncertainty quantification for BSG measures through functionals of model parameter posterior distributions via Bayesian estimation, compared to point-estimates from baseline VAR and NGC retrieval methods. We showcase how BSG can quantify strengths of temporal interactions (including spillovers) and identify systemically vulnerable nodes in a wildfire risk application.

We emphasize the distinction between Bayesian DAGs versus BSG, which models temporal, bi-directional relationships that can potentially amplify spillovers over multi-step horizons. DAG structure is a popular assumption in causal inference and can be viewed as a special case of BSG. BSG learns important edges (temporal interactions) and nodes (time series components) directly from estimated statistical network metrics. It also accounts for various dependencies in error terms that deviate from standard Gaussian noises, which are more descriptive of real-world systems. A brief overview of BSG vs. prior methods is shown in Figure 1.

## 2 METHODOLOGY

### 2.1 VECTOR AUTOREGRESSION (VAR)

Let $\mathbf{z}_t$ be a stationary $d$-dimensional multivariate time series, and $\{z_{jt}\}$ be the $j$-th component of this time series at time $t$. A VAR(p) model with order $p$ is defined as:

$$\mathbf{z}_t = \phi_0 + \sum_{i=1}^{p} \phi_i \mathbf{z}_{t-i} + \mathbf{a}_t \qquad (1)$$

where $\phi_0$ is a $d$-dimensional constant, $\phi_i$ is the $d \times d$ lag $i$ coefficient matrix for $i \geq 0$, and $\mathbf{a}_t$ is a sequence of i.i.d random vectors with mean 0 and covariance matrix $\Sigma_{\mathbf{a}}$.

**Bayesian Estimation.** We utilize a Bayesian approach [Tsay, 2013] for estimating unknown model parameters $[\boldsymbol{\beta}', \Sigma_{\mathbf{a}}]$ for a VAR(p) time series with length $T$, where $\boldsymbol{\beta}' = [\phi_0, \phi_1, ..., \phi_p]$:

$$\mathbf{Z} = \mathbf{X}\boldsymbol{\beta} + \mathbf{A} \qquad (2)$$

where $\mathbf{Z}$ and $\mathbf{A}$ are $(T-p) \times d$ matrices, and the $i$th row is $\mathbf{z}'_{p+i}$ and $\mathbf{a}'_{p+i}$. $\boldsymbol{\beta}'$ is a $d \times (dp+1)$ matrix, and $\mathbf{X}$ is a $(T-p) \times (dp+1)$ design matrix with $i$th row as $(1, \mathbf{z}'_{p+i-1}, \mathbf{z}'_i)$. The likelihood function for the data is

$$f(\mathbf{Z}|\boldsymbol{\beta}, \Sigma_{\mathbf{a}}) \propto |\Sigma_{\mathbf{a}}|^{-n/2} \exp[-\frac{1}{2} tr(\{(\mathbf{Z} - \mathbf{X}\boldsymbol{\beta})'(\mathbf{Z} - \mathbf{X}\boldsymbol{\beta})\Sigma_{\mathbf{a}}^{-1}\})] \qquad (3)$$

where $n = T - p$ is the effective sample size. We utilize Normal-inverse-Wishart conjugate priors $f(\boldsymbol{\beta}, \Sigma_{\mathbf{a}}) = f(\Sigma_{\mathbf{a}})f(\boldsymbol{\beta}|\Sigma_{\mathbf{a}})$ :

$$f(\Sigma_{\mathbf{a}}) \sim W^{-1}(\mathbf{V_0}, n_0) \qquad (4)$$

$$f(vec(\boldsymbol{\beta})|\Sigma_{\mathbf{a}}) \sim N(vec(\boldsymbol{\beta}_0), \Sigma_{\mathbf{a}} \otimes \mathbf{C}^{-1}) \qquad (5)$$

where hyperparameters $V_0$ is a $d \times d$ matrix, $n_0$ is some real number, $C$ is a $(dp+1) \times (dp+1)$ matrix, and $\beta_0$ is a $(dp+1) \times d$ matrix. The posterior distribution is then:

$$f(\Sigma_{\mathbf{a}}|\mathbf{Z}, \mathbf{X}) \sim W^{-1}(\mathbf{V_0} + \widetilde{\mathbf{S}}, n_0 + n) \qquad (6)$$

$$f(vec(\boldsymbol{\beta})|\mathbf{Z}, \mathbf{X}, \Sigma_{\mathbf{a}}) \sim N(vec(\widetilde{\boldsymbol{\beta}}), \Sigma_{\mathbf{a}} \otimes (\mathbf{X}'\mathbf{X} + \mathbf{C})^{-1}) \qquad (7)$$

where $\widetilde{\boldsymbol{\beta}} = ((\mathbf{X}'\mathbf{X} + \mathbf{C})^{-1}(\mathbf{X}'\mathbf{X}\widehat{\boldsymbol{\beta}} + \mathbf{C}\boldsymbol{\beta}_0))$ and $\widetilde{\mathbf{S}} = (\mathbf{Z} - \mathbf{X}\widetilde{\boldsymbol{\beta}})'(\mathbf{Z} - \mathbf{X}\widetilde{\boldsymbol{\beta}}) + (\widetilde{\boldsymbol{\beta}} - \boldsymbol{\beta}_0)'\mathbf{C}(\widetilde{\boldsymbol{\beta}} - \boldsymbol{\beta}_0)$ based on hyperparameter choices from the prior; $\widehat{\boldsymbol{\beta}}$ is the least-squares estimate of $\boldsymbol{\beta}$. Usually, $V_0$ is set to identity $\mathbf{I}_d$ and $n_0$ is a small number; as sample size $n$ increases, the choice of $n_0$ has very little effect on the final posterior. Similarly, we can choose vague priors for $vec(\boldsymbol{\beta})$ by letting $vec(\boldsymbol{\beta}_0) = 0$ and $\mathbf{C}^{-1} = c_0 I_{dp+1}$, where $c_0$ is some large real number, and hence the posterior distribution $f(vec(\boldsymbol{\beta})|\mathbf{Z}, \mathbf{X}, \Sigma_{\mathbf{a}})$ is also mainly updated via the data $\mathbf{X}$.

Although $\Sigma_{\mathbf{a}}$ is unknown, we can sample $M$ i.i.d samples from the joint posterior distribution by iterative sampling from $f(\Sigma_{\mathbf{a}}|\mathbf{Z}, \mathbf{X})$ and $f(vec(\boldsymbol{\beta})|\mathbf{Z}, \mathbf{X}, \Sigma_{\mathbf{a}})$, replacing $\Sigma_{\mathbf{a}}$ with posterior estimate $\Sigma_{\mathbf{a}}^{(m)}$.

### 2.2 BAYESIAN SPILLOVER GRAPHS

In brief, we adopt Bayesian estimation for Vector Autoregressions (VAR) to estimate posterior distribution for model parameters $[\boldsymbol{\beta}', \Sigma_{\mathbf{a}}]$ from a single realized MTS. We then construct $G_h(\boldsymbol{\beta}, \Sigma_{\mathbf{a}}|\mathbf{Z})$, the BSG for forecast horizon $h$, with components of MTS as nodes and temporal interactions as directed, weighted edges. Specifically, we can estimate BSG edge weights by computing $h$-step ahead normalized spillovers between two nodes via FEVD for $M$ posterior samples of $\{\boldsymbol{\beta}', \Sigma_{\mathbf{a}}\}$, and taking averages over $M$. Consequentially, BSG is an interpretable graph where both magnitude and specific values of edges are meaningful.

We also introduce three network measures based on functionals of BSG: the spillover index, vulnerability score, and influence score. These measures describe systemic-wide behavior over time and are useful for monitoring influential and at-risk nodes for a dynamic network. With a Bayesian framework, we can quantify uncertainty for both BSG edges and network measures. Under stationarity assumptions, estimated normalized spillovers are finite after some fixed forecast horizon $h$.

**Interpretable BSG Edges from Forcast Error Variance Decomposition.** We adapt generalized FEVD for analyzing $h$-step ahead spillover effects [Diebold and Yılmaz, 2014; Diebold and Yilmaz, 2015]; the accuracy of a forecast can be measured by its forecast error. Let $\sigma_{kk}$ be the k-th diagonal of $\Sigma_{\mathbf{a}}$, and $\psi_i$ be the coefficient matrix for a non-orthogonalized VAR under an infinite moving-average representation. The $jk$-th entry of the $h$-step ahead forecast error variance is

$$w_{h,jk} = \frac{\sigma_{kk}^{-1} \Sigma_{i=0}^{h-1} [\psi_i \Sigma_{\boldsymbol{a}}]_{jk}^2}{\Sigma_{i=0}^{h-1} [\psi_i \Sigma_{\boldsymbol{a}} \psi_i']_{jj}} \qquad (8)$$

which measures the amount of information of the $h$-step ahead forecast error variance for variable $j$ accounted for by innovations/exogenous shocks to variable $k$. The $h$-**step ahead normalized spillover** from component $k$ to $j$ is:

$$s_h^{k \to j} = 100 * \tilde{w}_{h,jk}, \quad \tilde{w}_{h,jk} = \frac{w_{h,jk}}{\Sigma_{k=1}^d w_{h,jk}} \qquad (9)$$

where $\tilde{w}_{h,jk}$ is the normalized variance decomposition. $s_h^{k \to j}$ is the proportion of the $h$-step ahead forecast error variance for node $j$ attributed to changes in node $k$, and becomes the weight for a directed edge from node $k$ to $j$ for BSG, $G_h(\beta, \Sigma_{\mathbf{a}}|\mathbf{Z})$. This definition makes BSG an interpretable graph with respect to forecast errors, with direct explanation of edge weight meaning. Prior methods such as GVAR would only estimate the sign of a temporal relationship [Marcinkevičs and Vogt, 2021]. See Algorithm 1 for details on estimating BSG edges from posterior distributions of Bayesian VAR parameters.

**BSG Network Measures as Systemic Risk Indicators.** We propose novel BSG network measures based on functionals of BSG edges over forecast horizon $h$ that can describe system-wide behavior and node importance over time. The goal is to quantify cumulative temporal interactions and spillovers within a system, as well as identify strongly influential or vulnerable nodes.

We define the **$h$-spillover index** as the magnitude of $h$-step normalized spillovers across all components, which describes the total spillover effect experienced over the full graph. The $h$-spillover index can be viewed as a measure of cumulative risk within the system after $h$ time periods; the higher it is, the more fragile the system is to innovations in any individual node.

$$S(\cdot) = S_h = \sum_{\substack{j=1 \\ j \neq k}}^{d} \sum_{k=1}^{d} s_h^{k \to j} \tag{10}$$

We may then be interested in identifying specific nodes at high risk over the full graph. For example, say we wanted to rank the individual nodes by the magnitude of spillovers experienced. We define $s_h^{* \to j}$ as the total spillover effect from all other components to a specific component $j$.

$$V(\cdot) = s_h^{* \to j} = \sum_{\forall k, k \neq j}^{d} s_h^{k \to j} \tag{11}$$

$s_h^{* \to j}$ can be viewed as the **vulnerability score** for a specific node at $h$-steps ahead, and can theoretically take on values between $[0, 100]$. The vulnerability score for node $j$ can be interpreted as the proportion of FEVD *not* attributed to innovations to $j$ itself. In particular, nodes with higher vulnerability are more susceptible to shocks and cascading effects from other components within the system.

Alternatively, we may be interested in pinpointing the sources of risks to the system. We define the **influence score** for a specific node, $s_h^{k \to *}$, as:

$$I(\cdot) = s_h^{k \to *} = \frac{\sum_{\forall j, j \neq k}^{d} s_h^{k \to j}}{S_h} \tag{12}$$

Note that the numerator of this expression quantifies the total spillover effect on the graph originating from component $k$, which is then standardized by the $h$-spillover index. This allows us to interpret the influence score for node $k$ as the proportion of total spillover effect on the entire system attributed to innovations in $k$, which again takes on values between $[0, 100]$ and is comparable across different networks. In particular, nodes with higher influence leads to greater impact on the entire system if there is a shock or change to the node. Collectively, these BSG network measures have wide applicability for describing real-world systems and as systemic risk indicators (SRI), which captures holistic risk arising from overall network connectivity [Che-Castaldo et al., 2021; De Bandt and Hartmann, 2000].

**BSG Estimation & Uncertainty Quantification.** Given a single realized MTS $\mathbf{Z_t}$, we can construct BSG $G_h(\boldsymbol{\beta}, \boldsymbol{\Sigma_a}|\mathbf{Z})$ directly via Bayesian VAR estimation. We first draw $M$ samples, $\{\beta^{(m)}, \boldsymbol{\Sigma_a}^{(m)}\}$, from the posterior distribution of model parameters. For fixed forecast horizon $h$, we compute $w_{h,jk}^{(m)}$, the $h$-step ahead forecast error variance, for each sample. BSG edges are then constructed by averaging over $M$, where $\bar{s}_h^{k \to j} = \frac{1}{M} \sum^M s_h^{(m),k \to j}$ is a weighted directed edge from node $k$ to node $j$. BSG nodes are the individual components of $\mathbf{Z_t}$. BSG network measures can also be computed directly by averaging over $M$ samples, e.g., the influence score for node $k$ would be estimated via $\bar{s}_h^{k \to *} = \frac{1}{M} \Sigma_{m=1}^{M} [\sum_{\forall j, j \neq k}^{d} s_h^{(m),k \to j} / S_h^{(m)}]$. See Algorithm 1. This process also allows for uncertainty quantification for any BSG edge or network measure by constructing credible intervals over $M$ estimates. We can also leverage the simplicity of Highest Posterior Density Interval (HPDI) or Bayes Factor [Kass and Raftery, 1995]. See Section 5 for an example with California wildfire data.

**Stationarity and Optimal $h^*$ for Equilibrium BSG.** A VAR(1) model can be written with an infinite sum as:

$$\mathbf{z}_t = \mu + \sum_{i=0}^{\infty} \psi_i a_{t-i} \tag{13}$$

where $\psi_i = \phi_1^i$ for $i \geq 0$ and $\mu$ is a $d$-dimensional constant. See Appendix A for details. If the series is **stationary**, then the absolute value of the eigenvalues of $\phi_1$ will be strictly less than 1. Various transformations, including detrending, removing seasonality, or differencing the series [Granger and Newbold, 2014] are recommended to ensure stationarity before parameter estimation. MTS with DAG temporal network structures can be viewed as a subset of VARs with restrictive assumptions on $\beta$. In the special case of a VAR(1) model where the temporal network structure of $z_t$ can be described by a DAG, $z_t$ is stationary; see Theorem 1 and proof in Appendix B.

**Theorem 1.** *If $\phi_1$ is a DAG, then (1) no component-wise autocorrelation exists, (2) $\phi_1$ can be specified by a strictly triangular matrix, (3) all eigenvalues of $\phi_1$ are 0 and hence $z_t$ is stationary.*

Under stationarity, BSG can reliably model cumulative response functions if shocks are not persistent and the system will return to equilibrium. See Algorithm 1 for choosing the optimal $h^*$-step. The horizon $h$ can be interpreted as a tuning parameter that controls the trade-off between learning immediate versus cumulative effects for BSG.

## 3 BSG FOR QUANTIFYING INDIRECT SPILLOVERS

We showcase how BSG models temporal spillovers that materialize after multiple periods. Consider a 5-dimensional

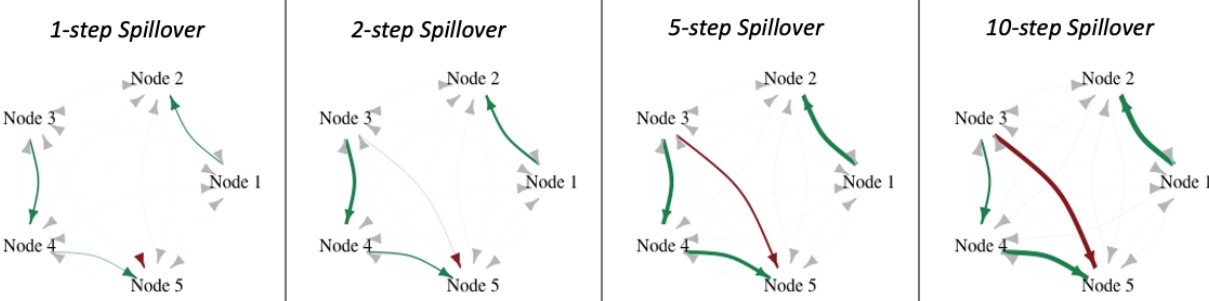

Figure 2: Normalized spillover evolution from Node 3 to 5 (red) over $h$. Arrow width is prop. to BSG edge strength.

Table 1: Average NDCG (Accuracy) for Identifying Sink & Source Nodes by Network Specification, 5 Rep.

| Stationary | 1. DAG, $d = 20$ | | 2. Directed Cyclic, $d = 20$ | | 3. Bipartite, $d = 20$ | |
|---|---|---|---|---|---|---|
| | NDCG@20 | NDCG@20 | NDCG@20 | NDCG@20 | NDCG@20 | NDCG@20 |
| Method | Source Nodes | Sink Nodes | Source Nodes | Sink Nodes | Source Nodes | Sink Nodes |
| BSG, $h = 1$ | $0.901 \pm 0.033$ | $0.997 \pm 0.004$ | $0.828 \pm 0.009$ | $1 \pm 0$ | $0.892 \pm 0.072$ | $0.988 \pm 0.009$ |
| BSG, $h = 5$ | $0.967 \pm 0.041$ | $\mathbf{0.998 \pm 0.002}$ | $\mathbf{0.959 \pm 0.039}$ | $\mathbf{0.999 \pm 0.001}$ | $\mathbf{1 \pm 0}$ | $\mathbf{1 \pm 0}$ |
| BSG, $h = 10$ | $\mathbf{0.966 \pm 0.041}$ | $\mathbf{0.998 \pm 0.002}$ | $0.962 \pm 0.037$ | $0.996 \pm 0.002$ | $1 \pm 0$ | $1 \pm 0$ |
| VAR-Between | $0.876 \pm 0.051$ | $0.722 \pm 0.051$ | $0.872 \pm 0.052$ | $0.726 \pm 0.052$ | $0.847 \pm 0.09$ | $0.702 \pm 0.09$ |
| VAR-Closeness | $0.79 \pm 0.042$ | $0.808 \pm 0.042$ | $0.785 \pm 0.069$ | $0.813 \pm 0.069$ | $0.76 \pm 0.08$ | $0.789 \pm 0.08$ |
| VAR-Degree | $0.936 \pm 0.034$ | $0.976 \pm 0.014$ | $0.931 \pm 0.037$ | $0.946 \pm 0.046$ | $0.981 \pm 0.033$ | $0.974 \pm 0.014$ |
| VAR-Eigen | $0.715 \pm 0.032$ | $0.883 \pm 0.032$ | $0.720 \pm 0.051$ | $0.879 \pm 0.051$ | $0.642 \pm 0.017$ | $0.908 \pm 0.017$ |
| DBN-Between | $0.766 \pm 0.047$ | $0.832 \pm 0.047$ | $0.766 \pm 0.044$ | $0.833 \pm 0.044$ | $0.674 \pm 0.078$ | $0.876 \pm 0.078$ |
| DBN-Closeness | $0.79 \pm 0.044$ | $0.809 \pm 0.044$ | $0.869 \pm 0.041$ | $0.729 \pm 0.041$ | $0.844 \pm 0.108$ | $0.705 \pm 0.108$ |
| DBN-Degree | $0.793 \pm 0.058$ | $0.827 \pm 0.038$ | $0.874 \pm 0.056$ | $0.855 \pm 0.053$ | $0.902 \pm 0.031$ | $0.858 \pm 0.071$ |
| DBN-Eigencentrality | $0.744 \pm 0.02$ | $0.854 \pm 0.02$ | $0.739 \pm 0.05$ | $0.859 \pm 0.05$ | $0.705 \pm 0.109$ | $0.845 \pm 0.109$ |
| GVAR-Between | $0.851 \pm 0.036$ | $0.747 \pm 0.036$ | $0.645 \pm 0.041$ | $0.954 \pm 0.041$ | $0.831 \pm 0.119$ | $0.719 \pm 0.119$ |
| GVAR-Closeness | $0.712 \pm 0.041$ | $0.886 \pm 0.041$ | $0.643 \pm 0.028$ | $0.955 \pm 0.028$ | $0.663 \pm 0.047$ | $0.887 \pm 0.047$ |
| GVAR-Degree | † | † | † | † | † | † |
| GVAR-Eigencentrality | $0.718 \pm 0.057$ | $0.881 \pm 0.057$ | $0.953 \pm 0.032$ | $0.646 \pm 0.032$ | $0.642 \pm 0.016$ | $0.907 \pm 0.016$ |

— indicates retrieved NGC graph is degenerate, e.g., only edges are self-directed.

† indicates network measure cannot distinguish between nodes, e.g., all in/out degrees are equal.

VAR(1) time series represented by the directed graph of temporal interactions ($\phi_1$) in Figure 3, with true parameters:

$$\phi_1 = \begin{bmatrix} 0.8 & 0.0 & 0.0 & 0.0 & 0.0 \\ \mathbf{0.5} & 0.8 & 0.0 & 0.0 & 0.0 \\ 0.0 & 0.0 & 0.8 & 0.0 & 0.0 \\ 0.0 & 0.0 & \mathbf{0.7} & 0.8 & 0.0 \\ 0.0 & 0.0 & 0.0 & \mathbf{0.4} & 0.8 \end{bmatrix} \quad (14)$$

$$\Sigma_a = diag(5). \quad (15)$$

Eigen-decomposition of $\phi_1$ indicates that all eigenvalues have magnitude $\leq 1$ and this network is stationary with standard independent error terms. Nodes 3 and 1 are analogous to source nodes with high out-degree centrality, and 5 and 3 to sink nodes with high in-degree centrality [Borgatti, 2005; Bollobás, 2012; Goldberg et al., 1989]. Node 5 will experience spillovers from Node 3 via Node 4 after multiple time periods, but this relationship is omitted in a simple NGC. This limitation is suitably addressed with a BSG with $h > 1$; see Figure 2 where indirect spillover (red arrow from 3 to 5) becomes stronger as $h$ increases.

In Figure 4, we plot average BSG directed edge weights ($h$-step ahead normalized spillover) from Nodes 1-4 into Node 5. The indirect spillover effect through intermediary Node 4 manifests after 2-steps ahead forecast and significantly amplifies as the forecast horizon increases (turquoise line) before flattening after $h = 17$. We can directly interpret this edge: the posterior mean for $s_{20}^{3 \to 5}$ is 80.1% with 95% HPDI of (71.9%, 87.7%), which predicts that after 20 periods, roughly 80.1% of forecast variability for node 5 can be attributed to changes in node 3. In contrast, the edge from Node 4 to Node 5 rapidly declines past $h = 4$. With prior methods of only estimating static NGC, we would not be able to observe nor quantify these spillover effects that evolve over longer forecast horizons.

## 4 BSG FOR IDENTIFYING NETWORK SOURCE & SINK NODES

We illustrate how BSG network measures accurately ranks and identifies nodes of interest compared to baselines with

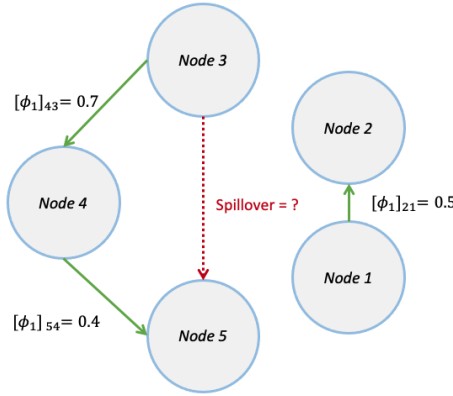

Figure 3: Graph of temporal interactions $\phi_1$ for a VAR(1) model. Goal is to quantify spillover effect over time (red).

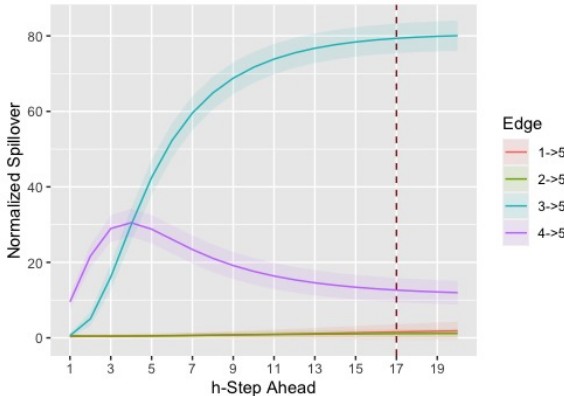

Figure 4: Edge strength (normalized spillover) into Node 5 over $h$. Direct impact via Node 4 (purple) declines over time while indirect spillover via Node 3 (turquoise) accumulates over time. BSG stabilizes at $h^* = 17$.

simulated MTS. Since relative order matters, this is a ranking instead of prediction task. Performance is evaluated by Normalized Discounted Cumulative Gain (NDCG) [Valizadegan et al., 2009]. NDCG measures ranking quality of a node ordering by BSG network measures or other graph measures, e.g., source nodes are ranked highly influential. NDCG is between $[0, 1]$ and directly comparable across methods; see Appendix C.

**Identifying Nodes Across Network Specifications.** 3 stationary network specifications ($\phi_1$) are used for simulating 5 MTS replicates: (1) a DAG, (2) a directed cyclic graph with autocorrelation = 0.5, and (3) a bi-partite graph. Networks (1) and (2) have 5 source and sink nodes and Network (3) has 10 source and sink nodes; all have independent Gaussian noise for $\Sigma_a$. Edge weights are sampled from a Unif(0,1) distribution; $T = 500$ and $d = 20$ for each network. We construct BSG[1] SRIs for $h = \{1, 5, 10\}$, and use influence and vulnerability scores for ranking source and sink

---

[1]Example code at https://github.com/gdeng96/bsg

**Algorithm 1** Estimating Bayesian Spillover Graph with Optimal $h^*$

Draw $M$ posterior samples for $\boldsymbol{\beta} = [\phi_0, \phi_1, ..., \phi_p]$, $\boldsymbol{\Sigma}_\mathbf{a}$
1: **while** $m < M$ **do** sample
2: $\quad \boldsymbol{\Sigma}_\mathbf{a}^{(m)} \sim W^{-1}(\mathbf{V_0} + \widetilde{\mathbf{S}}, n_0 + n)$
3: $\quad vec(\boldsymbol{\beta}^{(m)}) \sim N(vec(\widetilde{\boldsymbol{\beta}}), \boldsymbol{\Sigma}_\mathbf{a}^{(m)} \otimes (\mathbf{X'X} + \mathbf{C})^{-1})$
4: **end while**
   Iterate over $h$ until converge
5: **for** $h$ in 1, 2, ..., $H$ and $\epsilon > 0$ **do**
6: $\quad$ Compute $w_{h,jk}^{(m)}$ from $\boldsymbol{\Sigma}_\mathbf{a}^{(m)}, \boldsymbol{\beta}^{(m)}$
7: $\quad$ Compute $s_h^{(m),k \to j}$ from $w_{h,jk}^{(m)}$
8: $\quad$ Compute posterior mean $\bar{s}_h^{k \to j} = \frac{1}{M} \sum^M s_h^{(m),k \to j}$
9: $\quad$ **if** $|\bar{s}_h^{k \to j} - \bar{s}_{h-1}^{k \to j}| < \epsilon, \ \forall j, k$ **then**
10: $\quad\quad h^* = h$
11: $\quad$ **end if**
12: **end for**
    Construct BSG $G_h(\boldsymbol{\beta}, \boldsymbol{\Sigma}_\mathbf{a}|\mathbf{Z})$ with edges $\bar{s}_{h*}^{k \to j}$

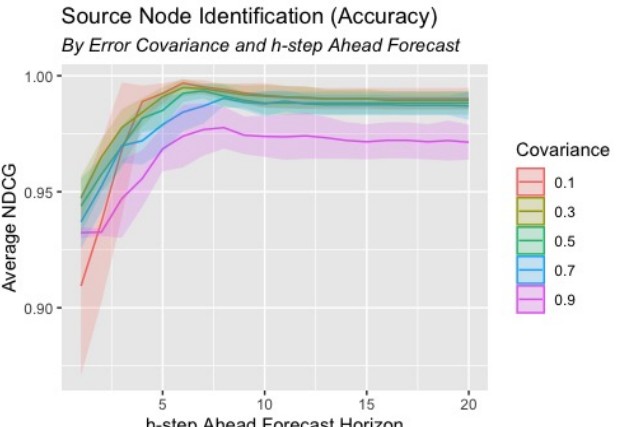

Figure 5: BSG Accuracy for identifying source nodes via influence scores, w.r.t. $h$-step ahead forecast horizon and different $\sigma_{jk}$ strengths.

nodes respectively. The first set of baselines are 4 standard graph measures on a NGC graph: in/out degree distributions, eigen-centrality, betweenness centrality, and closeness centrality. NGC is constructed from a VAR(1) model fitted via the MTS package, and significant edges are identified via multiple-testing with Benjamini-Hochberg procedure [Benjamini and Hochberg, 1995]. Another set of baselines is DBN and GVAR[2] combined with the 4 graph measures above, because these methods are designed only to retrieve NGC graphs. For fairness of comparison, GVAR lag is restricted to 1 and run with default hidden units/layer (50), hyperparameters $\lambda = 0.1$ and $\gamma = 0.01$, and 500 epochs in PyTorch. DBN uses default settings with the dbnR package.

Average NDCG are reported in Table 1 for each combina-

---

[2]GVAR code available at https://github.com/i6092467/GVAR

tion of baseline NGC graph-recovery method and network measure. Out- and in-degree centralities (Degree) are used for source and sink nodes respectively. BSG with $h = 10$ yields the highest accuracy for both node types across all three networks specifications.

**Effect of Forecast Horizon $h$ and Error Covariance $\Sigma_{\mathbf{a}}$**
We perform an ablation experiment to answer two questions: (1) *How does choice of hyper-parameter $h$ impact BSG quality and accuracy?* (2) *How well does BSG perform across different error dependency structures?*

We utilize Network (2), which allows for bi-directional temporal relationships and cycles. Each component has unit variance ($\sigma_{kk} = 1$), and pairwise covariance is $\{0.1, 0.3, 0.5, 0.7, 0.9\}$ corresponding to the strength of dependencies in $\Sigma_{\mathbf{a}}$. $d = 24$ with 8 source and sink nodes; for each $\Sigma_{\mathbf{a}}$ specification, we generate 5 replicates and estimate corresponding BSG for 20 values of $h$, then compute accuracy (NDCG) for source node identification. Figure 5 shows that good choices of $h$ ranges between 5-10, and BSG performance quickly stabilizes after a few forecast periods while successfully identifying the proper source nodes. Good choices for $h$ depends mostly on $\phi_1$ and is influenced by the speed at which the system reaches equilibrium (mean-reverts), not necessarily the size of the network. Lower $h$ values yield higher accuracy for identifying sink nodes; a good BSG should select $h$ that maximizes both quantities.

In Table 1 of Appendix D.1 , we report NDCG for identifying sink and source nodes in networks with weak, medium, and strongly correlated $\Sigma_{\mathbf{a}}$, using the same VAR, DBN, and GVAR specifications as previous experiments. Results show that BSG influence and vulnerability scores outperform all benchmarks even under strongly correlated error terms. When $\sigma_{jk}$ is moderately or strongly correlated, standard VAR breaks down and produces a degenerate graph (i.e., multiple testing results in zero significant edges); benchmark network measures collapse in this case. DBN performs mostly consistently, while for GVAR, corresponding in/out-degrees do not distinguish between influential nodes. BSG avoid these pitfalls since it inherently accounts for error dependencies and is more applicable for real-world dynamic networks with strong correlations.

**Non-Linear Dynamic Systems** Recent works have also focused on dynamic systems with non-linear or higher-order temporal relationships. A prime example is the Lokta-Volterra predator-prey model Bacaër [2011]. Four parameters $\{\alpha, \beta, \gamma, \delta\}$ correspond to prey $\rightarrow$ itself, predator $\rightarrow$ prey, predator $\rightarrow$ itself, and prey $\rightarrow$ predator interaction strengths. We generate 5 MTS replicates using the same parameter specifications ($\{1.2, 0.2, 1.1, 0.05\}$) as Marcinkevičs and Vogt [2021], with $T = \{50, 200, 1000\}$. We compare BSG influence/vulnerability scores vs. benchmarks for correctly identifying nodes as predator (source) and prey (sink). Results and example MTS simulation is reported in

Table 2 and Figure 1 in Appendix D.2; BSG at all forecast horizons outperforms baselines for $T = 50$ and $T = 200$. For $T = 1000$, BSG performs consistently well for identifying source nodes, but has lower accuracy for identifying sink nodes, likely due to long-range dependence for a longer MTS. GVAR-Closeness has marginally higher accuracy (+0.014) for identifying predators compared to BSG ($h = 1$) but very low accuracy (0.554) for identifying prey. Meanwhile, standard VAR after FDR adjustment produces degenerate graphs. On average, BSG still performs well on between both source and sink node identification; in practice, it may be useful to first difference MTS with higher-order autocorrelation.

# 5  BSG FOR UNDERSTANDING REAL-WORLD SYSTEMS

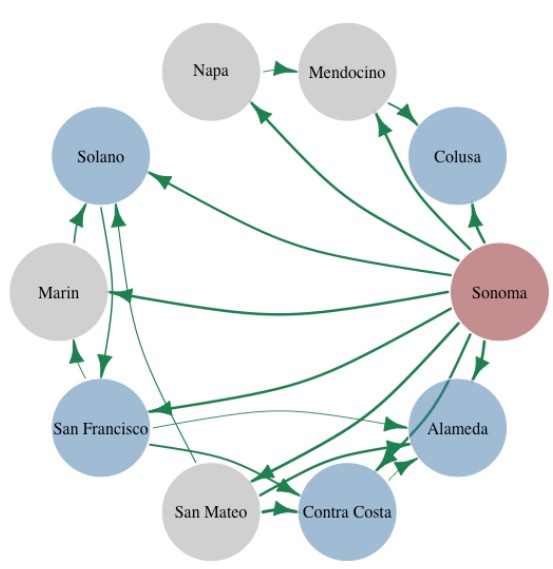

Figure 6: BSG for Kincade Fire, $h$=12 hours ahead. Red indicates source and blue indicates sink nodes. Arrow width is prop. to BSG edge weight. See Figure 4 in Appendix E for 95% HPDI of spillovers.

**Inferring Spillovers from California Wildfires.** The Kincade Fire was the largest California wildfire in 2019, burning a total of 77,758 acres. It originated in Sonoma County and dangerous PM10/PM2.5 particles in the air posed a serious public health risk spillover for nearby counties with high population density. We use BSG to investigate spillovers and rank at-risk nodes (counties) as measured by hourly PM 2.5 particle concentrations from Oct 22-Nov 7. We have a reasonable ground-truth for underlying network structure with Sonoma County as the single source node. Therefore, any strong BSG edges detected between Sonoma and non-adjacent counties, or two counties that does not include Sonoma, can be considered indirect spillover effects.

**Data Description.** Using public data from EPA (Environ-

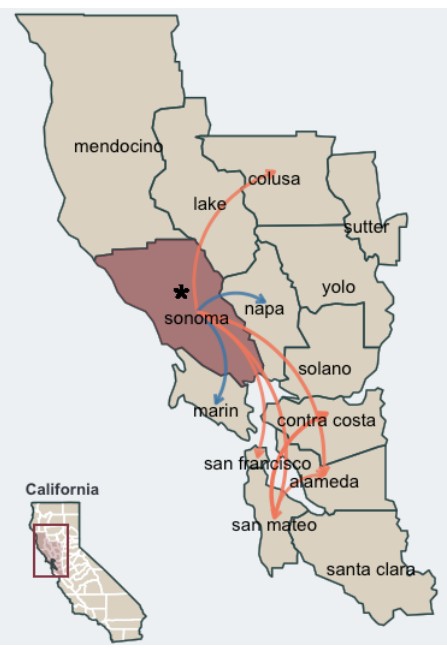

Figure 7: 12-hour normalized spillover for Kincade Fire. Blue arrows indicate direct risk for adjacent counties, and orange arrows indicate spillovers for non-adjacent counties.

mental Protection Agency), hourly PM 2.5 concentrations are extracted for 10 counties within 50 miles of Sonoma County in Northern California; Yolo, Sutter, and Lake counties had no data available. See Figure 2 in Appendix E for MTS plot. No visible trend or seasonality effects are observed; autocorrelation plots show evidence of long memory for some counties and we also observe prominent spikes, particularly initially in Sonoma and later with time lag in other counties. To ensure stationarity, we proceed with the first order difference of the MTS.

**Quantifying Spillover & At-risk Nodes.** In Figure 6, we illustrate all BSG edges ($h = 12$) greater than the 80th percentile in magnitude for simplicity, with arrow width proportional to edge weights. The top source node Sonoma (by BSG influence score) is shaded in red, and top sink nodes (by vulnerability score) is shaded in blue. The BSG neatly captures the Kincade Fire in that Sonoma has the majority of all outgoing edges, while further away, non-adjacent counties (sink nodes) such as Colusa and Alameda have strong spillovers both directly from Sonoma and indirectly via other counties as well. In particular, note the cycle from Sonoma → Contra Costa ↔ Alameda where sink nodes also interact and amplify spillover effects. We can further quantify downstream spillovers via BSG edge weights for counties to the southeast of Sonoma; see Figure 7 for county map with spillovers. Roughly 10% of FEVD for each county can be attributed to changes in Sonoma's PM 2.5 concentration. One possible explanation is downsloping winds from the north [Mass and Ovens, 2019], which is particularly

concerning due to the far higher population density of impacted counties. Two other notable indirect spillovers not involving Sonoma include those from San Mateo to Contra Costa (12.3%) and Alameda (9.3%).

BSG influence and vulnerability scores for each county are reported in Figure 3 in Appendix E. Sonoma County is the most influential node, accounting for more than 40.9% of total spillover effect across all 10 counties on average, with the 95% HPDI as (17.9%, 62.7%). BSG accurately identifies the origin of the Kincade Fire while also showing Sonoma itself is the least vulnerable node. Locations most at risk to the fire, by vulnerability score, are Alameda and Contra Costa followed by San Francisco, Solano, and Colusa. None of these 5 counties are adjacent to Sonoma; they incur higher risk via spillovers from intermediary Marin and Napa counties, accumulated over multiple time periods. These risk quantifications from BSG have practical implications for policies with respect to wildfire relief and public health. For example, although FEMA allocated nearly 60 million dollars in federal relief [FEM, 2019], the funds were strictly designated for Sonoma County. Meanwhile, BSG as an exploratory tool clearly identifies much broader spillovers and at-risk counties.

## 6  DISCUSSION

BSG is a novel framework for modeling temporal interactions and identifying important nodes within a dynamic system based on a single realized multivariate time series. BSG combines interpretable forecast error based network measures with uncertainty quantification via sampling from posterior graph distribution, and demonstrates robust performance across various graph specifications and error dependency structures. The hyperparameter $h$ allows for custom learning of both short and long-term temporal relationships, including indirect spillovers, which are better suited for understanding how real-world systems evolve over time. Careful choice of horizon $h$ can help model equilibrium state of systems and optimize proper ranking of sink and source nodes.

A key application of BSG could be for analyzing spillover impact in response to new regulations and economic policies. For example, consider when a significant event occurs in a particular city, e.g., a new tax policy is passed or a local manufacturer is shut-down and off-shored. Prior works have utilized impulse response functions to analyze policy interventions [Sims, 1980; Ericsson et al., 1998; Lütkepohl, 2005]; we propose leveraging BSG to examine and quantify both positive and negative externalities (spillover effects) in terms of employment statistics, traffic congestion, local rent, wages, etc., for neighboring cities or counties. Inference via BSG can be for both short-term and long-term impact based on forecast horizon, and used to inform both the public and policymakers.

Another potential BSG application is in time series analysis of fMRI data in healthcare and medicine [Penny et al., 2005]; for example, we can examine individual brain fMRI time series where each component are atlas based regions of interest, i.e. aggregated behavior from sets of voxels, which represent smaller unit regions in the brain. The time series could measure brain activity in response to some stimuli or treatment, and a BSG can illustrate cumulative effect of temporal interactions between different brain regions over time. The novel BSG network measures (influence score, vulnerability score) can also pinpoint critical components of brain connectivity, analogous to sink or source nodes.

Future work can dive deep into applying BSG for some of these datasets aforementioned, as well as extending the BSG framework for Bayesian networks with time-varying coefficients [Kowal et al., 2019] or latent state-space representations.

## Acknowledgements

The authors gratefully acknowledge financial support from the National Science Foundation Awards 1934985, 1940124, 1940276, and 2114143.

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
