# OpenReview forum: "Bayesian Spillover Graphs for Dynamic Networks"
_auai.org/UAI/2022/Conference — UAI 2022 Oral_

### Official Review · Reviewer_Rxup · 2022-04-11

**Q2(1) Originality/Novelty:** 3
**Q2(2) Significance/Impact:** 3
**Q2(3) Correctness/Technical Quality:** 3
**Q2(6) Clarity Of Writing:** 3
**Q6 Overall Score:** 7
**Q8 Confidence In Your Score:** 2

**Q1 Summary And Contributions:**

The paper introduces Bayesian spillover graphs, for learning temporal graphs. The main aim is to identify important nodes (sources/sink)
in a dynamic context based on a single time series. It combines forecast error variance decomposition (FEVD) with Bayesian VAR models to
quantify uncertainty.


**Q2 Assessment Of The Paper:**

More detailed information regarding each of these aspects is given below:

**Q2(4) Quality Of Experiments (Optional):**

4: Excellent: The experimental evaluation is comprehensive and the results are compelling.

**Q2(5) Reproducibility:**

3: Good: Key resources (e.g., proofs, code, data) are available and key details (e.g., proofs, experimental setup) are sufficiently well-described for competent researchers to confidently reproduce the main results.

**Q3 Main Strengths:**

1. Solid technical contribution: the combination of FEVD and Bayesian VARs is non-trivial. The resulting model is novel as far as I know and could have an impact in the analysis of real-world time series.
2. Extensive and convincing experiments with appropriate baselines.


**Q4 Main Weakness:**

The paper is not easy to follow, which makes it hard to pinpoint the exact contribution. I would have appreciated some more intuition about spillovers earlier on (it becomes clear later in the paper).


**Q5 Detailed Comments To The Authors:**

The proposed approach is an interesting contribution to network Granger causal modelling. The baseline approaches and evaluation metrics that are compared to seem appropriate.

I am not aware of similar work that focuses on spillover effects that are the focus of this paper. While I understood the intuitive notion later on in the paper, I think it would be useful to explain this briefly early in the paper.

The evaluations are extensive and provide some insight into the hyperparameter h. What I missed in the discussion was especially a discussion on the ablation experiments (the results of Figure 5). Is there an explanation for the fact that good choices for h are between 5 and 10? I also wondered if this should not strongly depend on the size of the network? Maybe this could be clarified in the rebuttal phase.

Minor comment/typo:
- Page 6; combinedd => combined.

**Q7 Justification For Your Score:**

Technically solid paper with good potential for impact.

**Q9 Complying With Reviewing Instructions:**

1: Yes.

---

### Official Review · Reviewer_rPjE · 2022-04-12

**Q2(1) Originality/Novelty:** 3
**Q2(2) Significance/Impact:** 4
**Q2(3) Correctness/Technical Quality:** 3
**Q2(6) Clarity Of Writing:** 4
**Q6 Overall Score:** 7
**Q8 Confidence In Your Score:** 4

**Q1 Summary And Contributions:**

In this paper, the authors proposed a new method for forecasting and other prediction analyses for multiple time series dynamic networks. Under simple assumptions, the authors were able to introduce three important network measures in determining spillover within a network through time. They then compare the result of forecasting using BSGs with other prior methods such as GCNs using synthetic data and prove its superiority.

**Q2 Assessment Of The Paper:**

More detailed information regarding each of these aspects is given below:

**Q2(4) Quality Of Experiments (Optional):**

3: Good: The experimental evaluation is adequate, and the results convincingly support the main claims.

**Q2(5) Reproducibility:**

3: Good: Key resources (e.g., proofs, code, data) are available and key details (e.g., proofs, experimental setup) are sufficiently well-described for competent researchers to confidently reproduce the main results.

**Q3 Main Strengths:**

1) The novelty of the method and its applicability in real-world scenarios especially in computational social sciences
2) Strong proof for algorithm and strong theoretical evidence proving the soundness of method derivation
3) Good simulations and comparison to prior methods


**Q4 Main Weakness:**

1) Lack of explanation on how this novel method can be applied in real-world scenarios other than a good example. At least a paragraph dedicated to how this method can solve societal or AI-related problems and can be applied can provide a lot of value.
2) Good references to other research overall, but references missing at times, especially when referring to “prior methods” for forecasting in dynamic networks. More explanation on prior methods, who derived them, what applications they have nowadays, and where they are being used in the AI world can be useful.
3) More explanation regarding formulae and the definition of some variables in the paper can be helpful.


**Q5 Detailed Comments To The Authors:**

Please provide a reference in the paper corresponding to this text in the abstract “...against state-of-the-art Bayesian Networks and deep-learning baselines.”. Please provide more information about this state-of-the-art and provide some references to the research papers.

Please provide a reference in the Introduction section for the following text. “Prior NGC methods also do not quantify strengths of temporal relationships nor provide ample interpretation for related graph measures”.

In Figure 1, for the “Previous Methods”, please update the text “Static NGC Graphs” to something along the lines of “Only applies to static NGC Graphs”. The red x next to the text might imply negation which is not the case here.
In Figure 1, for the “BSG” text in blue, please incorporate the word “dynamic” to better contrast with the word “static” for “Previous Methods”.

In equation 2, please provide an explanation of why the beta is being used in the equation but beta_prime is used in the explanation.

On page 3, in section 2.2, in the second paragraph where it is mentioned that three network measures are introduced, please list all three of them there.

On page 3, second text column, second line, there needs to be a period after the word “error”.

In equation 8, please define Psi and what it is referring to in the paragraph below.

In equation 9, please explain how the number 100 is being incorporated or provide a reference to where this equation is derived.

On page 4, second text column, first paragraph, line 6, the “a” at the end of the line should be removed.

On page 4, Section 3, first paragraph, line 3, “a directed” should be changed to “directed”.

Please provide the code and documentation for the “Inferring Spillovers from California Wildfires” experiment.


**Q7 Justification For Your Score:**

I leveraged the fact that this is a paper with a novel method for measuring spillover effects in dynamic systems which is superior to prior methods and the algorithms are well defined and well implemented and nice theoretical work has been performed. I also took into account the lack of some references and explanations and code/documentation.

**Q9 Complying With Reviewing Instructions:**

1: Yes.

---

### Official Review · Reviewer_aAr2 · 2022-04-12

**Q2(1) Originality/Novelty:** 2
**Q2(2) Significance/Impact:** 2
**Q2(3) Correctness/Technical Quality:** 3
**Q2(6) Clarity Of Writing:** 3
**Q6 Overall Score:** 7
**Q8 Confidence In Your Score:** 2

**Q1 Summary And Contributions:**

This paper proposes a novel framework, BSG, for modeling temporal interactions and identifying important nodes within a dynamic
system based on a single realized multivariate time series.

**Q2 Assessment Of The Paper:**

More detailed information regarding each of these aspects is given below:

**Q2(4) Quality Of Experiments (Optional):**

3: Good: The experimental evaluation is adequate, and the results convincingly support the main claims.

**Q2(5) Reproducibility:**

3: Good: Key resources (e.g., proofs, code, data) are available and key details (e.g., proofs, experimental setup) are sufficiently well-described for competent researchers to confidently reproduce the main results.

**Q3 Main Strengths:**

This model is fundamental and useful. I believe there should be many applications. This work already experiments on many.

**Q4 Main Weakness:**

I cannot find any clear weakness in this work.

**Q5 Detailed Comments To The Authors:**

The organization of this paper could be improved. The experiment part can be shrunk. Text and figures arrangement can be better.

**Q7 Justification For Your Score:**

Currently, I have no objective for this work. And I like this new model. I wish to see other reviewers' comments and to see if I will change my judgment.

**Q9 Complying With Reviewing Instructions:**

1: Yes.

---

### Decision · Program_Chairs · 2022-05-15

**Decision:**

Accept (Oral)

**Comment:**

Meta Review: In this paper, the authors proposed a new method for forecasting and other prediction  analyses for multiple time series dynamic networks. All the reviewers consider that the proposed method is fundamental and useful.